# Statistical methodologies for evaluation of the rate of persistence of Ebola virus in semen of male survivors in Sierra Leone

**Ndema Habib**[1]\*, **Michael D. Hughes**[2], **Nathalie Broutet**[1], **Anna Thorson**[1], **Philippe Gaillard**[1], **Sihem Landoulsi**[1], **Suzanne L. R. McDonald**[1], **Pierre Formenty**[3], **on behalf of Sierra Leone Ebola Virus Persistence Study Group**[¶]

1 UNDP/UNFPA/UNICEF/WHO/World Bank Special Programme of Research, Development and Research, Training in Human Reproduction, Department of Sexual and Reproductive Health and Research, World Health Organization, Geneva, Switzerland, 2 Department of Biostatistics, Harvard T.H Chan School of Public Health, Boston, Massachusetts, United States of America, 3 Department of Health Emergency Interventions, World Health Organization, Geneva, Switzerland

¶ Membership of the Sierra Leone Ebola Virus Persistence Study Group is provided in the Acknowledgments.
\* habibn@who.int

**Data Availability Statement:** Contractual agreements between the study parties (WHO/HRP,

## Abstract

The 2013–2016 Ebola virus (EBOV) outbreak in West Africa was the largest and most complex outbreak ever, with a total number of cases and deaths higher than in all previous EBOV outbreaks combined. The outbreak was characterized by rapid spread of the infection in nations that were weakly prepared to handle it. EBOV ribonucleic acid (RNA) is known to persist in body fluids following disease recovery, and studying this persistence is crucial for controlling such epidemics. Observational cohort studies investigating EBOV persistence in semen require following up recently recovered survivors of Ebola virus disease (EVD), from recruitment to the time when their semen tests negative for EBOV, the endpoint being time-to-event. Because recruitment of EVD survivors takes place weeks or months following disease recovery, the event of interest may have already occurred. Survival analysis methods are the best suited for the estimation of the virus persistence in body fluids but must account for left- and interval-censoring present in the data, which is a more complex problem than that of presence of right censoring alone. Using the Sierra Leone Ebola Virus Persistence Study, we discuss study design issues, endpoint of interest and statistical methodologies for interval- and right-censored non-parametric and parametric survival modelling. Using the data from 203 EVD recruited survivors, we illustrate the performance of five different survival models for estimation of persistence of EBOV in semen. The interval censored survival analytic methods produced more precise estimates of EBOV persistence in semen and were more representative of the source population than the right censored ones. The potential to apply these methods is enhanced by increased availability of statistical software to handle interval censored survival data. These methods may be applicable to diseases of a similar nature where persistence estimation of pathogens is of interest.

US CDC, China CDC, and the MoH Sierra Leone) exists and data rights reside with MOH-SL, the material owner. Inquiries related to the data, may be directed to Study Data Oversight Committee (SIS@who.int) with Subject: "Inquiry on Ebola Semen persistence study data" All essential study documents including analytic datasets will be stored in the HRP e-Archive system. They are not available for public access. We do not own individual level data, so upon request, and when appropriate data share agreements are in place, the data manager of the e-Archive system will be able to transfer the dataset to the recipient. If the de-identified study dataset has not been migrated to the e-Archive system yet, then the statistician or study data-manger would be sharing the dataset upon receipt of necessary data share agreements.

**Funding:** The Sierra Leone Ebola Virus Persistence Study (SLEVPS) team acknowledges the contributions of the WHO Ebola Response Program, the Paul G. Allen Family Foundation, and the UNDP (United Nations Development Program)–UNFPA (United Nations Population Fund)–UNICEF–WHO–World Bank Special Program of Research, Development and Research Training in Human Reproduction (HRP), a cosponsored program executed by the WHO; the US CDC, the China CDC, the Sierra Leone Ministry of Health and Sanitation and the Ministry of Defence, and the Joint United Nations Program on HIV/AIDS in support of the SLEVPS.

**Competing interests:** The authors have declared that no competing interests exist.

# Introduction

The 2013–2016 Ebola virus (EBOV) outbreak in West Africa, currently known as the largest and most complex outbreak since the virus was discovered in 1976, saw more cases and deaths than all earlier outbreaks combined [1]. Sierra Leone, Liberia and Guinea were the most affected countries. They contributed to the largest burden of Ebola virus disease (EVD) and deaths, with over 28,000 cases and over 10,000 EVD survivors requiring convalescent care [2]. The outbreak was marked by a rapid spread of infection in these three insufficiently prepared nations. It resulted in high case fatality rates (CFRs) reportedly 21.5%, 40.9%, and 60.8% in Sierra Leone, Liberia and Guinea respectively, and almost reversed developmental gains achieved over the previous years [3].

Following disease recovery, EBOV ribonucleic acid (RNA) has been detected in survivors various body fluids including sweat, saliva, urine and conjunctival fluid, with EBOV clearance in these body fluids occurring well under 100 days [4, 5]. However, studies show EBOV persists longer in semen [5, 6]. In the Sierra Leone Ebola Virus Persistence study (SLEVPS), Thorson et al. [6] reported a maximum duration of persistence of EBOV in semen of 696 days following discharge from Ebola treatment unit (ETU).

EBOV persistence in semen can be estimated by quantifying the risk (hazard) at which the virus clears from semen, which involves following up EVD survivors from disease recovery (after discharge from EVD treatment unit (ETU)) to the time when semen is confirmed to be negative for EBOV.

However, in EBOV persistence studies, time of EBOV clearance in body fluids cannot be observed with precision, either because the event occurred prior to first study visit, attributable to delays in recruitment, or between study visits. SLEVPS reported a median delay to recruit of 258 days (counted from ETU discharge) with 610 days as a maximum while the interval between scheduled consecutive visits for semen testing was two weeks [6, 7]. In Guinea's Post-EboGui study, a median delay from symptoms onset to recruitment was 319 days with a maximum of 810 days and the interval between two consecutive visits for semen testing ranged from 4–24 weeks [8].

Estimating EBOV persistence in semen is best implemented through application of survival analysis methods, due to the nature of the endpoint being time-to-event. An important advantage of these methods is their ability to handle data even when the survival time is not directly observed (or is censored).

There are three types of censoring encountered in survival. The first type which is the most encountered in prospective cohort studies in general is right censoring, whereby the event of interest has not yet occurred by the time of last visit. In the context of EBOV persistence, right censoring occurs when an EVD survivor who tested positive for semen on recruitment is yet to be confirmed EBOV-negative by the time of last contact, either because of their withdrawal from study or loss to follow-up (LFU).

The second type is left censoring whereby the event of interest has already occurred by the time of study recruitment however, with the interval during which the event occurred known. Left censoring is a common scenario in studies of EBOV persistence in body fluids and is caused by delayed entry (recruitment) of survivors at the time when the virus has already been cleared from the body fluid, with the interval in which this occurred known to be between ETU discharge and study recruitment [7, 9].

Left censoring is different from *left truncation* where the event of interest is not observed because the person was never enrolled in the study, for example, because they died before being enrolled. Left truncation is therefore *assumed* when participants whose event of interest occurred prior to recruitment are not included in a survival analysis.

The third type is *interval censoring* whereby the event of interest occurs within a specified time interval in the context of a periodic longitudinal study follow-up. The *interval censoring* can occur when the survivors who are EBOV-positive for semen on recruitment have the virus cleared in between follow-up visits. In studies of virus persistence in semen, it is common for the interval between visits for sample collection to be longer than planned. This may happen when a survivor cannot provide a semen sample during a scheduled study visit or when a sample is collected but does not meet the quality requirements for laboratory testing, necessitating a repeat sample collection at a later visit.

The date of earliest detection of EBOV in semen should theoretically be the starting point of observation in the estimation of the virus persistence in semen. However, this date is practically impossible to ascertain because of difficulties in obtaining semen samples from acute EVD patients for testing. On the other hand, understanding EBOV persistence during the post-acute infection period is of more public health interest in order to understand the possibility of sexual transmission of EBOV through semen.

Hence, in such studies, the population of interest is males who survived the acute EBOV infection phase, who would be expected to be sexually active again and therefore at risk of transmitting the virus. The survivors' date of discharge from ETU (following confirmed blood negative EBOV), in this case, serves as the starting point for estimating EBOV persistence in the semen. It has not been possible to collect semen samples for testing at the time of ETU discharge. However, the SLEVPS findings showed that the probability of EBOV-positivity for semen declined with increasing duration between the ETU discharge and recruitment; in various studies, it approached value of 1.0 with shorter duration [7, 10–12]. Based on SLEVPS, the assumption of EBOV-positivity for semen at ETU discharge seemed reasonable and was therefore assumed for this paper.

In epidemiology and public health, there has been a wide application of survival analysis methods dealing with right censoring [13–16]. In the context of a carefully designed clinical trial or any other study design in which the starting point of risk observation is fully under the control of the researcher, *left censoring* is expected to not to pose a problem, this being a more common scenario in public health. However, it is less common for the starting point of risk observation to be beyond the control of the researcher, like it is the case for EBOV persistence studies which requires utilization of appropriate methods to account for left censoring. The left- and right censoring are both special cases of interval censoring [17]. Currently, rich literature exists on the methods for analysis of interval censored outcomes, that include the use of non-parametric [18, 19], semi-parametric [20–22] and parametric methods [17, 23, 24]. There is also a handful of major statistical software for example SAS, R and STATA that are currently equipped with easy to apply survival routines to handle interval censored data [25–27]. But it proves occasionally necessary to use a combination of software, based on quality of graphical capabilities, and sometimes the need for manual computation of some parameters estimates whenever these cannot be directly obtained from the software. A single easy solution is not necessarily available, and a combination might be needed to overcome some limitations in available software.

Several studies have examined persistence of EBOV in body fluids, including semen, following clinical recovery from the disease, where maximum duration for virus positivity of the body fluid samples was reported [4, 28]. Sissoko *et al.*, [12] applied mathematical modelling of time-series viral load quantitative seminal fluid data threshold cycle (Ct) of 26 EVD survivors in a cohort study setting to systematically determine the dynamics of virus persistence over time, and using the model predicted median and 90th percentile times for virus clearance. However, there was no indication of how the authors accounted for the interval censored nature of the data in the time-series modelling.

There is limited literature illustrating how the right- and interval censored survival techniques can be applied in the estimation of persistence of EBOV in body fluids, given the study design. From the review of current literature, only one paper, by Subtil *et al.*, [8], was identified that reported follow-up and persistence of EBOV in semen among 188 male EVD survivors (Guinea PostEboGui study), and applied survival methodologies that accounted for the interval censored nature of the data. However, there was no thorough description of how the determination of the lower and upper bounds of the left- and interval censored events was implemented.

This paper is aimed at describing the theoretic, study design and methodological considerations for non-parametric and parametric survival approaches for estimating persistence of Ebola virus in semen in the presence of interval censoring. Using SLEVPS design, the paper illustrates the application of these methodologies; discusses the resulting persistence estimates from different models; and highlights strengths and weaknesses of each of these approaches for EBOV persistence estimation in semen.

## Materials and methods

### Sierra Leone Ebola virus persistence study: Aims, population, design and data collection procedures

SLEVPS recruitment took place from May 2015 to May 2016 in Sierra Leone in two locations: the 34 Military Hospital (MH34) (an urban facility in Freetown, Western District) and Lungi Government hospital (a semi-rural facility in Lungi, Port Loko District). EVD survivors were recruited through meetings held in collaboration with the Sierra Leone Association of Ebola Survivors, and other survivor support groups.

The study consisted of a convenience sample of 220 adult male survivors of EVD, enrolled in two phases, at various times after discharge from an ETU. The survivors were followed prospectively to determine the duration and correlates of persistence of EBOV in semen. Eligible consenting survivors provided semen specimens at recruitment and two weeks later (the two baseline visits). Those specimens were tested for the presence of EBOV RNA using a quantitative reverse transcriptase polymerase chain reaction (qRT-PCR) test. Follow-up visits continued until semen tested twice consecutively qRT-PCR negative for EBOV RNA.

The qRT-PCR test targeted two genes for EBOV detection in semen: NP and VP40 during phase 1 of the study, and NP and GP in phase 2 of the study [7]. For the persistence analysis purposes using survival methods, the semen specimen was considered EBOV-*positive* if there was a detection of EBOV RNA in one or both gene targets; and EBOV-*negative* if there was no detection of EBOV RNA in both gene targets. *Confirmed EBOV negativity* occurred when there were two consecutive EBOV-negative results from semen specimens collected at any two consecutive visits.

Those found to be EBOV-positive for any of the two baseline specimens were followed-up every two weeks thereafter until the semen specimens tested EBOV-negative on two consecutive visits. EBOV-positive or -negative semen test results were considered as *valid* results, whereas non-interpretable EBOV results (due to semen specimen poor quality, insufficient quantity or contamination) were considered as *non-valid* and therefore excluded from the persistence analysis.

The primary event of interest was confirmed EBOV negativity (EBOV clearance) in semen with the endpoint being the time to confirmed EBOV negativity in semen, measured in days from the date of ETU discharge. The date of confirmed EBOV negativity was the earlier of two consecutive dates with samples showing EBOV-negativity in semen. The date of ETU

discharge was chosen as the time of origin (Time zero) due to interest in persistence during the post-recovery period for EBOV disease.

For this study, right censoring was implemented at the visit prior to the last to ensure *independent (non-informative) censoring*, which is an important assumption in analyzing censored survival data [29, 30]. The earliest opportunity for study staff to collect and test a semen specimen was at the first recruitment (baseline) visit.

The study population, implementation, specimen collection and testing, as well as the nature of the collected baseline social, clinical and behavioural indicators during and after the EVD acute phase have been thoroughly detailed elsewhere [7, 9].

## Ethics

Ethical permission was granted from the Sierra Leone Ethics and Scientific Review Committee and the WHO Ethical Review Committee (No. RPC736). All study participants signed an informed consent.

## Primary outcome assessment, study participant types and design considerations

Fig 1 illustrates different time points ($t_1$, $t_2$ and $t_3$, measured in days) of assessment of confirmed EBOV negativity status in semen, as determined from the date of ETU discharge (time zero), for three types of SLEVPS participants ($P_1$, $P_2$ and $P_3$) grouped according to whether they experienced the event of interest, and in case they did, by when this was observed. It was assumed that all the recruited participants were EBOV-positive in semen at time zero.

Let $t_1$ be the time from ETU discharge to study entry (recruitment) visit for the participants who had a *valid* EBOV semen test result at this point. For those who *did not have a valid* EBOV semen test result, $t_1$ becomes the time from ETU discharge to the first visit beyond recruitment having a *valid* EBOV semen test result.

$P_1$ are those participants who became confirmed EBOV-negative for semen at time $t_1$ and are therefore considered as left censored. $P_2$ would be those who were EBOV-positive for semen at $t_1$ and became confirmed EBOV-negative during study follow-up at time $t_2$. On the other hand, $P_3$ would be those participants who were EBOV-positive for semen at time $t_1$ and became right censored at time $t_3$.

Two types of study populations are in consideration: *Population $S_0$*, that includes all recruited EVD survivors, independent of the status of the event of interest at time $t_1$; and *Population $S_1$*, a sub-population of $S_0$ that includes only survivors who were yet to experience the event of interest by time $t_1$ (includes $P_2$ and $P_3$ only). Population $S_1$ is used in this paper to illustrate the biases associated with assuming left-truncation (exclusion) of observations of participants $P_1$.

## Survival analysis methods for persistence estimation

We have chosen for illustration interval censored survival methods that correctly treat persistence data as interval censored; and for comparison, included the right censored survival methods that ignore the interval censored nature of the persistence data. For the interval censored survival methods, we illustrate how the persistence is estimated using the non-parametric survival methods as well as the parametric methods which assume the distribution of the persistence data is known.

**The right censored survival approaches.** The right censored (RC) survival analysis approaches are standard methods commonly applied when the time of occurrence of an event observed is known exactly or is right censored. Because the exact time at which the event

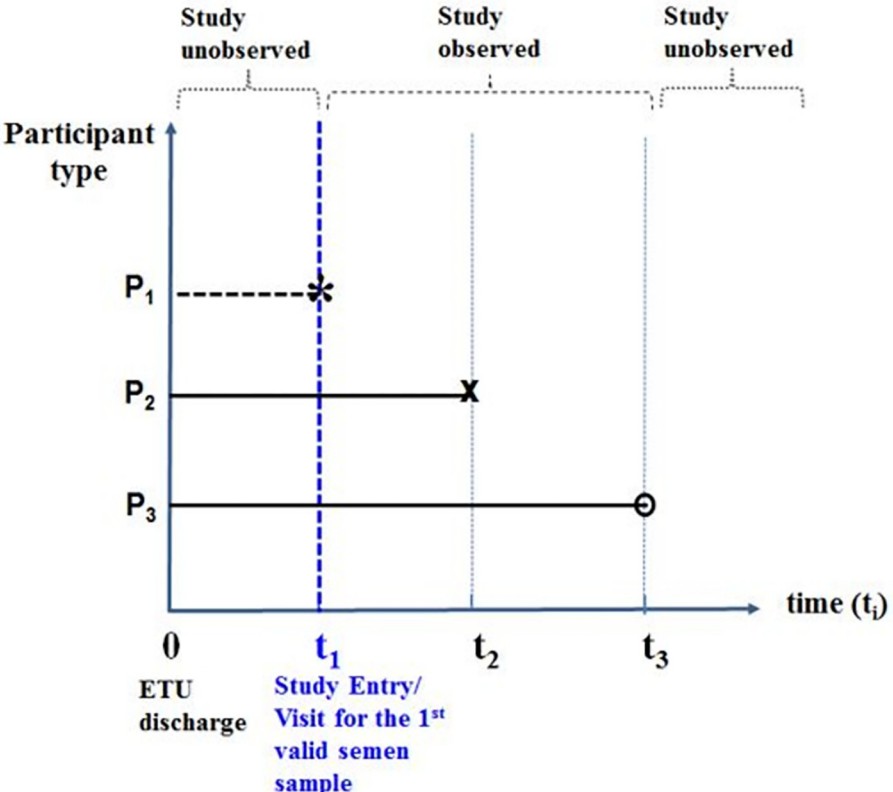

**X** Event (confirmed EBOV negative in semen after first baseline visit)
**O** Right-censored (EBOV-positive at study discontinuation or lost to follow-up)
**✻** Left-censored (confirmed EBOV-negative at or earlier than at $t_1$ visit)

**Fig 1. Study participants time to confirmed negative Ebola virus RNA in semen, by type of censoring experienced.**

occurs cannot always be observed for endpoints which can only be observed at regular intervals of visits, the right censoring methods can still be applied by assuming the time of event as equal to the time of the visit at which the event is first diagnosed as having occurred, or by imputing the time of event at the midpoint of the interval between the last visit at which the event is yet to occur and the visit at which the event is first diagnosed.

Let T denote a random variable for time duration (in days) between the date of ETU discharge and the date of reaching confirmed EBOV negativity in semen. Let δ be a censoring indicator at the observed time points ($t_1$, $t_2$ and $t_3$) with value set to 1 if the participant is confirmed negative for EBOV in semen; or set to 0 otherwise. The following two approaches can be used to assign values for T and δ for the right censored survival models, with and without assuming left truncation of observations:

### Approach 1: Assigning value of T as equal to time from ETU discharge to the first observed confirmed EBOV-negativity and assuming left truncation of the observations for participants of type $P_1$

When left truncation is assumed, the participants will be included for persistence analysis conditional on being confirmed negative later than at time $t_1$ hence use of population $S_1$. The

**Table 1. Right censored survival methods: The time duration from ETU discharge to confirmed EBOV negativity ($T_i$) and the censoring status ($\delta_i$) for populations $S_0$ and $S_1$, and by type of participants.**

| Participant type | Time from ETU Discharge to visit with event[a], observed or to the last visit (t) | Approach 1 | | Approach 2 | | Approach 3 | |
|---|---|---|---|---|---|---|---|
| | | Time to first time, event of interest[†] observed, assuming left truncation for $P_1$ | | Time to first time, event of interest[†] observed assuming event of interest occurred at time $t_1$ for $P_1$ | | Time to mid-point between last time, event[†] not observed and first time, event[†] observed | |
| | | Sub-population $S_1$ | | Population $S_0$ | | Population $S_0$ | |
| | | Endpoint T | Censoring indicator δ | Endpoint T | Censoring indicator δ | Endpoint T | Censoring indicator δ |
| $P_1$ | $t_1$ | Excluded | Left truncated | $T = t_1$ | 1 | $T = \frac{t_1}{2}$ | 1 |
| $P_2$ | $t_2$ | $T = t_2$ | 1 | $T = t_2$ | 1 | $T = \frac{t_2 + l_2{}^b}{2}$ | 1 |
| $P_3$ | $t_3$ | $T = t_3$ | 0 | $T = t_3$ | 0 | $T = t_3$ | 0 |

[a] Event of interest = confirmed EBOV-negativity in semen.

[b] Value $l_2$ (not shown in Fig 2) is directly retrieved from the data, as the time of the last EBOV-positive result (prior to time $t_2$) for type $P_2$ participant.

values of (T, δ) for $P_2$ and $P_3$ are ($t_2$, 1) and ($t_3$, 0) respectively (Table 1, Approach 1). The limitation of using this population is reduced sample size due to the left-truncation of $P_1$ observations and therefore decreased efficiency of the model parameter estimates because of not using all available data. Furthermore, population $S_1$ may not be representative of the population where the Ebola virus disease survivors originated, as it favours inclusion for analysis of those with prolonged EBOV persistence (became confirmed negative beyond time $t_1$) over their peers in terms of duration $t_1$ from ETU with shorter EBOV persistence (became confirmed negative earlier than at time $t_1$). This therefore biases the results towards longer persistence duration.

## Approach 2: Assigning value of T as equal to the time from ETU discharge to earliest observed confirmed EBOV-negativity

This is Population $S_0$ which includes all the recruited EVD survivors ($P_1$, $P_2$ and $P_3$). By including participants $P_1$ in this population under the right censoring survival techniques, one must assume that they became confirmed EBOV-negative at time $t_1$. The values of (T, δ) for $P_1$, $P_2$ and $P_3$ are ($t_1$, 1), ($t_2$, 1) and ($t_3$, 0) respectively (Table 1, Approach 2).

The advantage of using this population is increased sample size, by using data for all recruited survivors. The main weakness however is increased likelihood of overestimation of the overall persistence rate and duration, by ignoring the likelihood that confirmed EBOV negativity in semen among $P_1$ participants may have occurred earlier than at time $t_1$.

## Approach 3: Applying single imputation of time to event, with T equal to the time to the mid-point between visits for the last EBOV-positive and the first confirmed EBOV-negative result, as counted from ETU discharge

Because the time of event is not always directly observable, estimation of event time, by use of single imputation using the midpoint of the interval between two visits is a commonly applied approach that enables application of right censored survival models in the presence of interval censored data [31–33].

Specific to SLEVPS, for participant $P_1$ the imputed time duration for T can be estimated as equal to $\frac{t_1}{2}$. For participant $P_2$, this is estimated as the duration to the midpoint between two consecutive time points: $l_2$ -the time of the latest visit at which the participant was observed to

be still EBOV-positive, and $t_2$—the time of the visit at which he was observed as confirmed EBOV negative for the first time, which equals $\frac{l_2+t_2}{2}$. For participant $P_3$ the censoring time T is equal to $t_3$ because their observations have been right censored. In this case the values of (T, δ) for $P_1$, $P_2$ and $P_3$ are $(\frac{t_1}{2}, 1)$, $(\frac{l_2+t_2}{2}, 1)$ and $(t_3, 0)$ respectively (Table 1, Approach 3).

The main limitation of the mid-point imputation approach is that the persistence estimates obtained may be less accurate, especially if the interval duration from ETU discharge to time $t_1$ varies widely between participants of type $P_1$. For SLEVPS, this interval ranged from 4 to 9 months [7]. It has been reported that using the midpoint of an interval for estimation of time at which the event occurs, can lead to biased effect estimates [31, 34]. The midpoint approach may furthermore underestimate standard errors, especially when the intervals are wide and of varying length [35].

With values of T and δ in the format as shown in Table 1 for the right censored survival approaches 1–3, a non-parametric maximum likelihood estimator (NPMLE) right censored Kaplan-Meier (KM) estimator [36] can be used to estimate EBOV persistence rate in semen.

The KM (product-limit) estimator for persistence at time t, S(t), for right censored survival will be defined as $\hat{S}(t) = \begin{cases} 1 & if \quad t < t_{1*} \\ \prod_{t_i \leq t}\left[1 - \dfrac{d_i}{Y_i}\right] & if \quad t_{1*} < t \end{cases}$ where $t_{1*}$ represents the first (observed or imputed) time of the confirmed EBOV-negativity event (failure time), counting from ETU discharge; with $d_i$ the number of survivors confirmed to be EBOV-negative; and $Y_i$ the number of those not yet confirmed negative and have not been censored, by time t.

**The interval censored survival approaches.** Under the interval censored (IC) approach, the exact time T of confirmed negativity for EBOV will be contained in an interval between two time points (L, R], where L is defined as the latest time at which the participant was observed or known to be still EBOV-positive and R as the earliest time at which he was observed as confirmed EBOV-negative. For the left censored participants, L will be the time at ETU discharge (Time 0) and R will be the time $t_1$. For the right censored participants, L will be at the visit at time $t_3$ and R can be set to infinity ($\infty$). In majority of statistical programs, the infinite value of R for the right censored individuals is usually set to missing. For the participants whose confirmed EBOV-negativity occurred between two study visits, their time T is considered as interval censored.

Table 2 shows the respective interval censoring intervals for the three types of participants $P_1$, $P_2$ and $P_3$ who were left-, interval- and right censored, being equal to (0, $t_1$], ($l_2$, $t_2$] and ($t_3$, $\infty$) respectively. To apply this approach the lower and upper limits of the interval (L, R] such that L< T ≤ R have to be determined.

Approaches 4 and 5 below show how the interval censored non-parametric and parametric survival approaches can be applied to estimate EBOV persistence, with the persistence data put in the format (L, R].

## Approach 4: The non-parametric maximum likelihood estimator Kaplan Meier's Turnbull interval censored model

The non-parametric maximum likelihood estimator (NPMLE) is one of developments implemented in the statistical analysis programs that permit use of the non-parametric KM methods to analyze interval censored data. Consider a sample of *n* subjects from a homogeneous population of male EVD survivors followed from ETU discharge to confirmed EBOV-negativity in semen and having non-informative interval censored observations $\{I_i\}_{i=1}^n = \{I_1, I_2, \ldots, I_n\}$ where $I_i = (L_i, R_i]$ is the interval known to contain the unobserved T for the $i^{\text{th}}$ subject.

**Table 2. Interval censoring methods: Distribution of the lower and the upper limits of censoring interval at which the failure time of interest, T occurred, for the three scenarios, based on population $S_0$.**

| Participant type | Time from ETU discharge to visit with event[a] of interest observed or to the last visit (t) | INTERVAL CENSORED APPROACHES KM NPMLE Survival (Approach 4) and Parametric Survival (Approach 5) Population $S_0$ | | |
|---|---|---|---|---|
| | | L | R | Type of censoring |
| $P_1$ | $t_1$ | $L = 0$ | $R = t_1$ | Left censoring |
| $P_2$ | $t_2$ | $L = l_2{}^b$ | $R = t_2$ | Interval censoring |
| $P_3$ | $t_3$ | $L = t_3$ | $R = \infty$ | Right censoring |

[a] Event of interest = confirmed EBOV-negativity in semen.

[b] Value $l_2$ (not shown in Fig 2) is directly retrieved from the data, as the time of the last EBOV-positive result (prior to time $t_2$) for type $P_2$ participant.

From the observed $\{I_i\}_{i=1}^n$, a set of non-overlapping intervals $\{(p_j, q_j]\}_{j=1}^m$ where

$p_1 \le q_1 < p_2 \le q_2 < p_3 \le q_3 < \cdots < p_m \le q_m$ is generated, over which the non-parametric EBOV persistence rate function $S(t) = P(T_i > t)$ is estimated.

Let $\alpha_{ij}$ denote the event indicator in which it is equal to 1 if the interval $(p_j, q_j] \subseteq I_i$ and equals to zero otherwise. Let $\vartheta_j = S(p_j) - S(q_j)$ be the weight in the $j^{th}$ interval and the probability of a confirmed EBOV-negativity event occurring in this interval.

Assuming independence, the vector parameter $\vartheta = (\vartheta_1, \vartheta_2, \ldots, \vartheta_m)'$ can be estimated by maximising with respect to $\vartheta_1, \vartheta_2, \ldots, \vartheta_m$ the likelihood

$L_S(\vartheta) = \prod_{i=1}^n Prob\{L_i < T_i \le R_i\} = \prod_{i=1}^n [S(L_i) - S(R_i)] = \prod_{i=1}^n \sum_{j=1}^m \alpha_{ij} \vartheta_j$, under the condition that $\sum_{j=1}^m \vartheta_j = 1$ and $\vartheta_j \ge 0$ for j = {1, 2, . . ., m} [18]. One of the algorithms that can be used to maximize $L_S(\vartheta)$ is an Expected Maximization Iterative Convex Minorant (EM-ICM) algorithm [37].

The maximum likelihood estimates (MLEs) $\vartheta_1, \vartheta_2, \ldots, \vartheta_m$ would yield the NPMLE of EBOV persistence function S(t) to be uniquely determined over observed non-overlapping intervals

$(p_j, q_j]$, and given by $S(t) = \begin{cases} 1 & if\ t < q_1 \\ \sum_{k=j+1}^m \hat{\vartheta}_k & if\ p_j \le t \le q_{j+1} \\ 0 & t > p_m \end{cases}$

SAS Procedure ICLIFETEST, with a built-in capability for interval censored data [38], available in SAS/STAT Version 14.1 [26] can be used to estimate the KM interval censored NPMLEs of the EBOV persistence rate in semen. This procedure applies the EM-ICM algorithm that supports the Turnbull algorithm [18] and computes standard errors using multiple imputation methods. SAS Procedure ICLIFETEST uses by the default 1000 multiple imputations. The EBOV persistence rate estimates obtained from this model are available only in a set of non-overlapping intervals and cannot be uniquely estimated in the case of overlapping (Turnbull) intervals between participants. Other major statistical analysis software which can also provide the NPMLEs of the interval censored data, include R packages "Interval" [27, 39] or "icenReg" with call function *ic_np* (where *np* stands for *non-parametric*)or relatively large samples with >100,000 observations [27, 40]; and also STATA "IntCens" package [25, 41, 42].

## Approach 5: Interval censored Weibull (parametric) model

One advantage of parametric models is that they tend to give more precise parameter estimates when there is a good fit to the data, since they are based on fewer parameters compared to the non-parametric survival models. Exponential, log-normal, log-logistic and Weibull are among the commonly used parametric survival distributions. For this paper we chose the Weibull distribution in *apriori*, because of its flexibility as both a proportional hazard (PH) as well as an accelerated failure time (AFT) model; and furthermore because it estimates and forecasts more accurately with extremely small samples.

The Weibull model can be fitted for the interval censored data in the (L, R] format, with or without baseline covariates. For this study, the Weibull persistence probabilities were estimated based on expected times (from ETU discharge) to EBOV clearance, using the estimated the Weibull shape parameter given as $\alpha = {}^1/_\sigma$ (where $\sigma$, is the extreme value scale parameter estimate) and scale $\lambda = \exp(\mu)$ (where $\mu$ is the intercept parameter estimate) obtained from the fit of an intercept-only model in SAS Procedure LIFEREG. Hence the semen EBOV persistence survival curve using Weibull distribution can expressed in terms of the scale $\lambda$ and shape $\alpha$ as follows: $S(t; \lambda, \alpha) = \exp(-({}^t/_\lambda)^\alpha)$ [26], whereby shape $\alpha$ gives an indication of whether the hazard rate, in this case rate of confirmed EBOV negativity in semen, decreases ($\alpha < 1$), is constant ($\alpha = 1$) or increases ($\alpha > 1$) over time: while scale $\lambda > 0$, determines the duration of persistence of EBOV in semen. There is also an alternative parameterization of the Weibull survival function which can also expressed as $S(t; b, \alpha) = \exp(-bt^{-\alpha})$ where scale $b$ is expressed as $b = \lambda^{-\alpha}$. For this paper we used the earlier parameterization.

In addition to SAS, other statistical packages that can fit Weibull and other parametric survival models to interval censored survival-time data include R using function "survreg" [27]; and STATA package "stintreg" [41–43].

We used the SLEVPS data to illustrate the estimation of persistence of EBOV in semen using the five survival models. SAS software was used for estimation of median EBOV persistence duration and the corresponding 95% confidence interval (CI). We used R statistical software Version 3.1 to plot EBOV persistence curves emanating from the estimates produced by the five approaches. For plotting the interval censored KM persistence curve in R, "Icens" and "Interval" packages were used [44], with the "Icens" package implementing an Expected-Maximization (EM) algorithm to obtain the survival estimates.

**Estimation of percentiles of EBOV persistence and 95% confidence interval.**   *Percentiles*. Let the p$^{th}$ percentile, denoted as $t_p$ (where $p = \{50, 75, 90\}$) represent the smallest observed time following ETU discharge at which probability of EBOV persistence in semen, $S(t_p) < (1 - p/100)$. The values of $t_p$ were estimated directly from the survival functions of the five models with: SAS Procedure LIFETEST for non-parametric EBOV persistence estimation assuming data is right censored; ICLIFETEST procedure used for the non-parametric estimation assuming the data is interval censored; and LIFEREG procedure for parametric estimation assuming Weibull-distributed interval censored EBOV persistence data.

*Standard errors for the percentiles*. The standard errors (SE) of $t_p$ were estimated following the methodology outlined in the book by Collett [45].

Let $t_{(j)}$ be the j$^{th}$ ordered confirmed EBOV-negativity event time (j = 1, 2, . . ., r).

The SEs for the four non-parametric EBOV persistence KM models (Approaches 1–4) were computed as follows:

$SE\left(t_p\right) = \frac{1}{\hat{f}(t_p)} \times SE\{\hat{S}(t_p)\}$, where $\hat{f}\left(t_p\right) = \frac{\hat{S}(\hat{u}_p) - \hat{S}(\hat{l}_p)}{\hat{l}_p - \hat{u}_p}$; with

$\hat{u}_p = Max\left\{t_{(j)} | S\left(t_{(j)}\right) \geq \left[1 - \left(\frac{p}{100}\right) + \epsilon\right]\right\}$ as the maximum observed time where KM estimate of EBOV persistence probability $\geq \left[1 - \left(\frac{p}{100}\right) + \epsilon\right]$; and

$$\hat{l}_p = Min\left\{t_{(j)}|S\left(t_{(j)}\right) \leq \left[1 - \left(\frac{p}{100}\right) - \epsilon\right]\right\}$$ as the smallest observed time $t_{(j)}$ where KM esti-mate of EBOV persistence probability $\leq \left[1 - \left(\frac{p}{100}\right) - \epsilon\right]$. The value of $\epsilon = 0.05$ was used.

The values of $\hat{u}_p, \hat{l}_p, \hat{S}(\hat{u}_p), \hat{S}(\hat{l}_p)$ and $SE\{\hat{S}(t_p)\}$ were obtained from the SAS output of the KM survival models. SAS-estimated $SE\{\hat{S}(t_p)\}$ using Greenwood formula and imputed SEs were used for $SE\{\hat{S}(t_p)\}$ for the KM-RC and KM-IC models respectively. Following directly from above, the corresponding lower and upper confidence limits of $t_p$ for the four right- and interval censored KM models were estimated linearly as $t_p \mp 1.96 \times SE(t_p)$.

The SE of $t_p$ for the Weibull parametric interval censored model (Approach 5) was directly invoked from SAS Procedure LIFEREG. The lower and upper 95% confidence limits of the percentiles given by the formula [45] $\exp\left[\ln\left(t_p\right) \mp \{1.96 \times SE\left(\ln(t_p)\right)\}\right]$ where $SE\left(\ln(\hat{t}_p)\right) = \frac{1}{t_p} \times SE\left(\hat{t}_p\right)$; with the $t_p$ and $SE(t_p)$ values.

## Results

Table 3 shows the distribution of survivors entering intervals of follow-up (in days) relative *to time point $t_1$* and the corresponding number of survivors who became confirmed EBOV-nega-tive during each of the intervals. This table shows that 88 out of the 203 participants recruited at time $t_1$, were already confirmed EBOV-negative by this time ($P_1$ participants).

Fig 2 illustrates survival curves for the five candidate approaches used for estimation of EBOV persistence in semen. The KM right censoring (Approach 1) which assumes the con-firmed negativity occurred at the first time it is observed and in addition assumes left trunca-tion for $P_1$ participants, results in the persistence curve that is shifted to the right, leading to overestimation of EBOV persistence duration.

The KM right censored (Approach 2) which also assumes the confirmed negativity occurred at the first time it is observed, and at time $t_1$ for $P_1$ participants, also results in an overestimation of persistence which is more extreme than that in Approach 1. Survival models applying KM right censored midpoint imputation (Approach 3), KM interval censored multi-ple imputations (Approach 4) and Weibull interval censoring (Approach 5), yield persistence curves which are much closer together and persistence rate estimates which are much lower compared to those from Approaches 1 and 2. The fit of the Weibull model on the EBOV per-sistence data yielded the scale ($\lambda$) parameter value of 251.6 (95% CI 230.1, 275.1) days. It also yielded a shape ($\alpha$) parameter value of 2.14 (95% CI 1.84, 2.49), which is above 1.0, indicating

**Table 3. Crude follow-up time (in days) and observed confirmed EBOV status of male survivors counting from enrolment visit $t_1$[a].**

| Start time interval (days) from enrolment visit ($t_1$) | # entering the interval | # withdrawn from study | # confirmed negative for EBOV |
|---|---|---|---|
| **Enrolment (visit $t_1$[a])** | 203 | 0 | 88 |
| **1–30** | 115 | 1 | 42 |
| **31–60** | 72 | 1 | 19 |
| **61–90** | 52 | 1 | 11 |
| **91–180** | 40 | 0 | 28 |
| **181–270** | 12 | 1 | 5 |
| **271–360** | 6 | 2 | 3 |
| **361–450** | 1 | 0 | 0 |
| **451–540** | 1 | 1 | 0 |
| **Total # with at least one semen specimen with valid results** | | 7 | 196 |

[a] $t_1$ refers to the recruitment or post-recruitment visit at which the semen sample collected yielded the first valid (positive or negative) result for EBOV result.

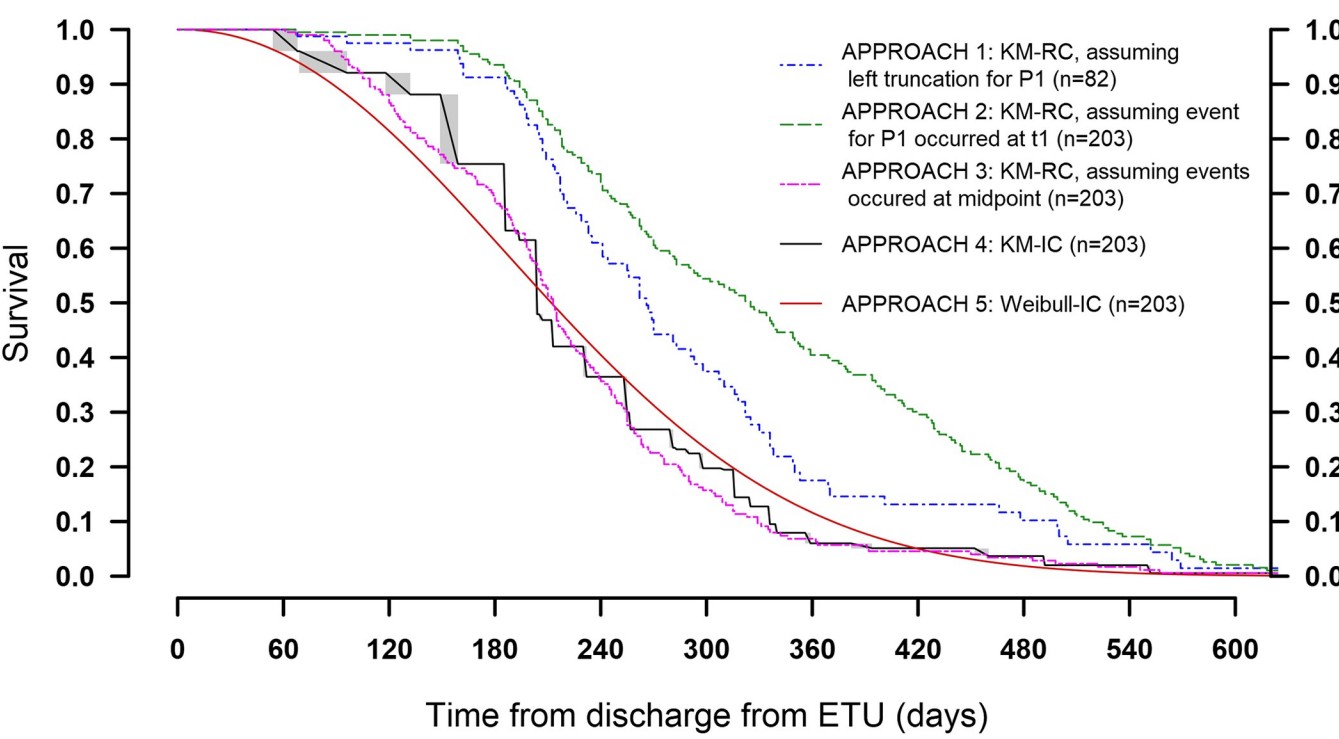

**Fig 2. EBOV persistence using right- and interval censored non-parametric and parametric approaches.**

rate of clearing of the virus in the semen increases with time, consistent with the observed SLEVPS persistence data. When the KM-IC and Weibull persistence curves are plotted together their 95% confidence intervals clearly overlapped (S1 Fig).

Fig 3 shows that the 50[th], 75[th] and 90[th] percentiles (95%CI) for EBOV persistence in semen of the EVD survivors indicating the respective times at which persistence probability was below 0.50, 0.25 and 0.10 respectively. KM IC model (Method 4) shows the persistence probability (95%CI) was <0.50 at 204 (193, 215) days, < 0.25 at 281 (244, 318) days, and was under 0.10 at 336 (300, 372) days post-ETU discharge. Approaches 3 and 5 that took into consideration the interval in which the event occurred, produced percentile estimates which were much closer to those obtained through KM IC model. Approaches 1 and 2 which did not take into account event interval produced percentiles which deviated substantially from those of KM IC model.

## Discussion

The non-parametric and parametric survival models applying the right and interval censoring methodologies presented in this paper illustrated differing results in the estimation of EBOV persistence in semen. The point estimates for the rate and duration of EBOV persistence in semen as well as their precision as obtained from these models varied considerably. The right censoring survival methods that assume the confirmed negativity occurred at the first time it is observed (Approaches 1 and 2) resulted into persistence curves which were more shifted to the right towards higher persistence rate and longer persistence duration. The median duration of

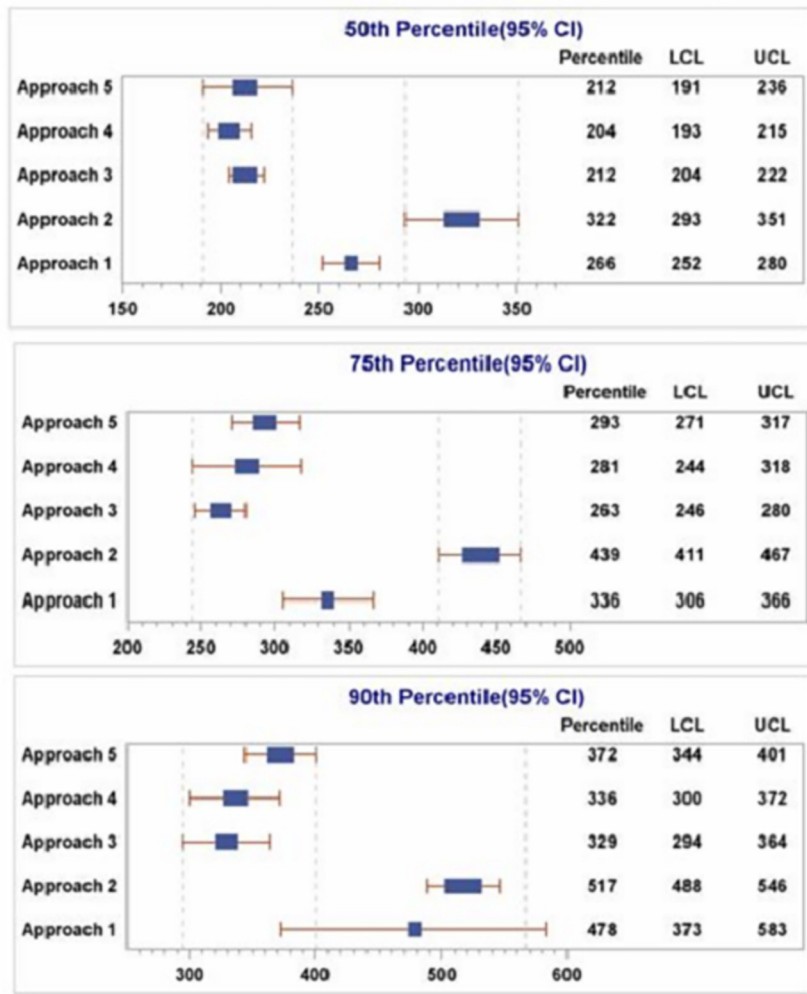

Approach 1: KM RC assuming all events occurred at R; and left truncation for $P_1$
Approach 2: KM RC assuming all events occurred at R, with events for $P_1$ occurring at $R=t_1$
Approach 3: KM RC assuming all events occurred at mid-point of the interval (L, R]
Approach 4: KM_IC (L, R]
Approach 5: Weibull IC (L, R]

**Fig 3. Comparison of the performance of the five non-parametric and parametric models in estimating percentiles (95%CI) for EBOV confirmed negativity in semen.**

EBOV persistence using these two approaches was shown to be about 2–4 months longer compared to KM-IC method (Approach 4). Approaches 1 and 2 resulted in 75th and 90th percentile estimates which further deviated from those of KM-IC method (higher by 4–6 months) and produced the least precise estimates of the 50th, 75th and 90th percentiles of the persistence curve (Fig 3). On the other hand, the right censored method that applied a single midpoint imputation of the time (Approach 3) fared comparatively better, in terms of yielding estimates of persistence rate and duration that were comparable to those obtained using the interval censored approaches. This method also resulted in a more precise median EBOV duration, consistent with the KM-IC method.

The results of the EVD survivors' data show that the Weibull IC EBOV persistence curve when considered relative to the KM-IC curve, fitted each other well beyond 400 days post-ETU discharge, with the point estimates for persistence rate for the Weibull curve slightly lower or above those of the KM-IC curve in the period before and after 200 days post-ETU, respectively (Fig 2). The Weibull IC distribution however produced estimates of EBOV persistence in semen that were almost comparable to those of KM-IC model.

It has been reported that using right censoring survival analysis methods to analyze data that consists of left- or interval censored observations may result into biased estimates, and severely underestimated standard errors [46].

Left censoring was present in the SLEVPS with 88 (43%) of 203 participants confirmed EBOV negative on recruitment. This was also reported in the Guinea's PostEboGui study by Subtil *et al*., [8] where 173 (91.9%) out of the 188 male EVD survivors tested negative for EBOV in semen on recruitment following discharge from the treatment centre whereby both parametric and non-parametric (Turnbull) estimators were used in the persistence estimation.

Relative to the EBOV persistence in semen estimation, three types of biases may have been induced because of applying the single imputation right censored KM survival models.

The first type is selection bias due to left-truncation of the observations of participants confirmed EBOV-free in semen at the time of first specimen with valid result was obtained (Visit $t_1$), (Approach 1). This bias leads to loss of sample information since the participants excluded at this time who had a shorter EBOV persistence duration might be characteristically different from their included peers who had longer persistence (beyond time $t_1$) despite both groups being recruited at around the same time from ETU discharge. Furthermore, there is loss of sample size which would affect the precision of the persistence endpoint estimates.

The second type of potential bias is due to failure to consider the time interval during which the confirmed EBOV negativity occurred in the survival analysis (Approaches 1 and 2). The magnitude of this bias is dependent on how long the interval is between visits containing time at which the event occurs. This however is important for the Sierra Leone cohort since some EVD survivors had a long interval between visits. Firstly, there was a long interval from when they were discharged from ETU to the time they were recruited, where for a vast majority of the participants this period was longer than 3 months and went as high as 19 months. The effect of this is seen in Approach 2, since the inclusion of the left censored participants by imputing their time at which the event occurs at $t_1$ led to a shift of the persistence curve to longer durations of persistence. This shift was more extreme in this study even relative to the Approach 1 which applies the same methodology but truncates the observations for the left censored participants. The right censoring survival model with the single midpoint imputation (Approach 3) is also prone to this type of bias especially when the intervals between visits are too long.

The third possible bias may be as a result of possible underestimation of standard errors due to single imputation of the right censored survival methods. However, from the results, the right censored KM model with midpoint imputation resulted in median duration estimates and precision which did not deviate much from those obtained through the KM-IC model.

The KM-IC model hence is the most appealing for estimating EBOV persistence in semen as it is the most efficient and does not require prior distributional assumptions for the baseline hazard.

Several major statistical software packages that can handle interval censored proportional hazards regression modelling that account for covariates adjustment. For the Sierra Leone study, SAS Procedure ICPHREG with a piecewise constant parameterization for the baseline hazard was used to fit an interval censored proportional hazards (PH) regression model that explored and adjusted for important predictors and effect-modifiers of being EBOV-free in

semen [6]. Other statistical software that integrate covariates in the semi-parametric regression model include R package "icenReg" with call function "ic_sp" (where sp stands for semi-parametric) [28, 41] and STATA package "stintcox" [42, 43]. Fully parametric interval censored multivariable regression models can be fitted also using SAS Procedure LIFEREG; R package "incenReg" call function "ic_par"; and STATA package "stinreg".

Percentiles of virus persistence in semen provide the probability of EBOV persisting beyond a certain time period. This is of clinical and public health importance as it helps with informing semen testing survivor programmes and policy formation surrounding duration of use of certain preventive measures (including sexual abstinence and condom use aimed at minimizing sexual transmission of the virus), and therefore the possibility of preventing future outbreaks. Furthermore, extreme upper tail virus persistence percentiles are important in understanding duration following ETU discharge that a group of survivors who are slowest to clear EBOV, become EBOV-negative.

One challenge faced was in the estimation of the SEs for the lower and upper tails of the non-parametric (KM) survival percentile distributions. While current statistical procedures like SAS ICLIFETEST or LIFETEST can easily estimate the SEs for the central survival percentiles ($25^{th}$, $50^{th}$ and $75^{th}$) also referred to as survival quartiles, these routines do not automatically estimate the SEs for the extreme lower and upper percentiles. For consistency, the standard errors for the $50^{th}$, $75^{th}$ and $90^{th}$ percentiles for the EBOV persistence in this paper were computed manually using the formulae outlined in the book by Collett [29], combined with SAS-produced estimates required in these respective formulae. The 95% confidence limits for the survival quartiles computed manually were compared against those readily estimable in the SAS program and showed a difference in width of the intervals between the two methods of estimation of the percentiles of the 4 non-parametric (KM) models (under linearly transformed 95% CI) not exceeding three weeks. For the Weibull interval censored model, the LIFEREG procedure had the in-built capability to estimate all the percentiles and corresponding SEs.

## Conclusions

Survival models that take into account the interval nature of the data on EBOV persistence in semen ensure statistically robust and unbiased estimates of EBOV persistence in this body fluid. Through comparison of estimates obtained using the right and interval censoring approaches, the methodologies that account for interval censoring result in shorter confidence interval (and therefore more precise estimates) which are also more representative of the source population compared to right censored approach (that ignore interval censoring). With increasing availability of statistical routines like SAS, R, STATA and other software to handle interval censored data, it has become relatively easier to apply them. The non-parametric and semi-parametric interval censoring survival methods should therefore be highly considered for use in estimation of virus persistence in body fluids of EVD survivors. Where good fit is demonstrated, the parametric interval censored methods including those that use the Weibull distribution should be considered as they give more precise estimates. These models can also be applied to study persistence in other types of pathogen such as Zika virus.

## Supporting information

**S1 Fig. EBOV persistence: Comparison of interval-censored Weibull model to the KM model with 95% CI.** "Republished from [Thorson AE, Deen GF, Bernstein KT, Liu WJ, Yamba F, Habib N, et al. (2021) Persistence of Ebola virus in semen among Ebola virus disease survivors in Sierra Leone: A cohort study of frequency, duration, and risk factors. PLoS Med

## Acknowledgments

Dr Gilda Piaggio, Statistician, Geneva, Switzerland
Dr Soe Soe Thwin, Statistician, The World Health Organization

## Sierra Leone Ebola Virus Persistence Study (SLEVPS) Group

**Sierra Leone Ministry of Health and Sanitation**: Gibrilla Fadlu Deen (principal investigator), James Bangura, Amara Jambai, Faustine James, Alie Wurie, Francis Yamba.
**Sierra Leone Ministry of Defence**: Foday Sahr, Thomas A. Massaquoi, Foday R. Sesay,
**Sierra Leone Ministry of Social Welfare, Gender, Children's Affairs**: Tina Davies,
**World Health Organization**: Nathalie Broutet (Principal Investigator), Pierre Formenty, Anna E. Thorson, Archchun Ariyarajah, Florence Baingana, Marylin Carino, Antoine Coursier, Kara N. Durski, Faiqa Ebrahim, Ndema Habib, Philippe Gaillard, Margaret O. Lamunu, Sihem Landoulsi, Jaclyn E. Marrinan, Suzanna L. R McDonald, Dhamari Naidoo, Carmen Valle, Teodora Wi, Zabulon Yoti.
**United States Centers for Disease Control and Prevention**: Barbara Knust (principal investigator), Neetu Abad, Aneesh Akbar-Uqdah, Sarah D. Bennett, Kyle T. Bernstein, Aaron C. Brault, Bobbie Rae Erickson, Elizabeth Ervin, Sara Hersey, Jill Huppert, John D. Klena, Tasneem Malik, Oliver Morgan, Dianna Ng, Stuart T. Nichol, Lydia Poroman, Lance Presser, Christine Ross, Tara K. Sealy, Ute StroÈher,
**Chinese Center for Disease Control and Prevention**: Wenbo Xu (principal investigator), Mifang Liang, Hongtu Liu, William Jun Liu, Guizhen Wu, Yong Zhang,
**Joint United Nations Programme on HIV/AIDS (UNAIDS)**: Patricia Ongpin.

## Author Contributions

**Conceptualization:** Ndema Habib, Michael D. Hughes, Nathalie Broutet, Anna Thorson, Philippe Gaillard, Pierre Formenty.

**Data curation:** Ndema Habib, Sihem Landoulsi.

**Formal analysis:** Ndema Habib.

**Investigation:** Ndema Habib, Nathalie Broutet, Anna Thorson, Philippe Gaillard, Sihem Landoulsi, Suzanne L. R. McDonald, Pierre Formenty.

**Methodology:** Ndema Habib, Michael D. Hughes, Nathalie Broutet, Anna Thorson, Suzanne L. R. McDonald, Pierre Formenty.

**Project administration:** Nathalie Broutet, Anna Thorson, Philippe Gaillard.

**Resources:** Ndema Habib, Sihem Landoulsi, Suzanne L. R. McDonald.

**Software:** Ndema Habib, Sihem Landoulsi.

**Supervision:** Philippe Gaillard, Suzanne L. R. McDonald.

**Validation:** Ndema Habib, Suzanne L. R. McDonald, Pierre Formenty.

**Visualization:** Ndema Habib.

**Writing – original draft:** Ndema Habib.

**Writing – review & editing:** Ndema Habib, Michael D. Hughes, Nathalie Broutet, Anna Thorson, Philippe Gaillard, Sihem Landoulsi, Suzanne L. R. McDonald, Pierre Formenty.

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
