## [Decision Letter · Decision Letter 0]

7 Apr 2021

PONE-D-21-05351

Statistical methodologies for evaluation of the rate of persistence of Ebola virus in semen of male survivors in Sierra Leone

PLOS ONE

Dear Dr. Habib,

Thank you for submitting your manuscript to PLOS ONE. After careful consideration, we feel that it has merit but does not fully meet PLOS ONE’s publication criteria as it currently stands. Therefore, we invite you to submit a revised version of the manuscript that addresses the points raised during the review process.

We look forward to receiving your revised manuscript.

Kind regards,

Mohammad Asghari Jafarabadi

Academic Editor

PLOS ONE

2. Thank you for conducting your study according to STROBE guidelines. We ask that you complete and upload a copy of the STROBE checklist (http://www.strobe-statement.org) as a supplemental file.

3. We noted in your submission details that a portion of your manuscript may have been presented or published elsewhere.

"Yes. Figure 3 has been published before in Thorson A. et al. et al. (2021) Persistence of Ebola virus in semen among Ebola virus disease survivors in Sierra Leone: A cohort study of frequency, duration, and risk factors. PLoS Med 18(2): e1003273. https://doi.org/10.1371/journal.pmed.1003273. This figure has been used because it displays the comparison of estimates of interval-censored non-parametric survival model to those of parametric Weibull model. It further shows the overlap in the 95% confidence interval for the two models."

Please clarify whether this publication was peer-reviewed and formally published. If this work was previously peer-reviewed and published, in the cover letter please provide the reason that this work does not constitute dual publication and should be included in the current manuscript.

5. One of the noted authors is a group or consortium [Sierra Leone Ebola Virus Persistence Study Group]. In addition to naming the author group and listing the individual authors and affiliations within this group in the acknowledgments section of your manuscript, please also indicate clearly a lead author for this group along with a contact email address.

Reviewers' comments:

Reviewer's Responses to Questions

**Comments to the Author**

1. Is the manuscript technically sound, and do the data support the conclusions?

Reviewer #1: Partly

Reviewer #2: Partly

2. Has the statistical analysis been performed appropriately and rigorously? 

Reviewer #1: Yes

Reviewer #2: Yes

3. Have the authors made all data underlying the findings in their manuscript fully available?

Reviewer #1: No

Reviewer #2: No

4. Is the manuscript presented in an intelligible fashion and written in standard English?

Reviewer #1: Yes

Reviewer #2: Yes

5. Review Comments to the Author

Reviewer #1: The authors of this manuscript present convincing evidence for use of statistical methods that consider interval censoring when examining persistence of EBOV in semen among SLEVPS participants. Their methods section does a good job introducing different statistical options along with their weaknesses. However, the writing in other sections of this paper, while grammatically correct, is often not written with clear arguments and supporting evidence. The introduction and discussion sections suffer in particular. It is unclear whether the authors are writing a methods tutorial paper with SLEVPS as a convenient example, if they are writing specifically to researchers who work with data like SLEVPS, or if they are trying to improve estimates of EBOV persistence in semen. I recommend that the authors think about what they are trying to communicate specifically with this manuscript and to whom. Once that is identified, I think it will be easier to resolve many of the issues. Full comments in attached file.

Reviewer #2: The authors present an interesting paper that describes different approaches to analysis of time to event data subject to different kinds of censoring. This endeavour deserves merit, and they explain very nicely the different approaches and show their results using a data set with information about traces of the Ebola virus in semen.

However, there are several issues that have to be addressed before I can recommend a publication of the manuscript:

Major issues:

1. It is unclear what the main aim of using these models is. Should individual out-of-sample predictions be possible in order to advise future Ebola survivors? Or does the main interest lie in observing the mean or median behaviour (or the percentiles) of the population of Ebola survivors?

2. This point is closely related to the former point and actually the most important part of my review: The authors can only describe the different estimated survival curves/percentiles/standard errors. But it is impossible to say which of the models is best, because the authors fail to define criteria which can be used to measure the model quality. At several places the authors claim that one or the other model is unbiased, but they cannot say this without a criterion for model quality. They sometimes describe that a model leads to more precise estimates in the sense of having smaller confidence intervals, but this does not necessarily mean that the respective model is correct.

Depending on the main goal of fitting these models (see point 1), I suggest the following changes: If the main goal is prediction, the authors should predefine criteria for predictive model choice such as proper scoring rules (e.g. the Brier score) that can take into account both sharpness (precision) and calibration of a predictive model or judge the calibration alone. In addition they should separate the data set in two parts, the training set and the test set, because otherwise the model fit will always be too good. Alternatively they could use some kind of cross-validation.

If, however, the main aim of the models is parameter estimation/a description of the population, they should show some criteria for model fit/model choice such as some version of AIC/BIC/DIC or the likes that are appropriate for the models they use and allow for a somewhat independent comparison of the quality of the models. In addition, they could add a plot that shows the observed data, so that it is easier to judge.

If the authors don't want or cannot do this, they could alternatively give a restructured description of the different approaches and their advantages and disadvantages (e.g. distributional assumptions, number of parameters,...) and only show the results of the models as an addition at the and, but they should remove all judgement from the results part that is not justified (all about bias or the notion that a more precise confidence interval is always better).

3. It becomes already clear from the description of the approaches that approach 1 and 2 that take only right-censoring into account are not appropriate for the data set. Therefore it is not at all surprising that the estimates are quite far away from the ones from the models for interval-censored data, this statement is a bit trivial. It is still nice to see, but it does not add much to the paper, while using relatively much space in the methods part. I recommend to shorten the explanation a bit and focus more on the other three methods.

4. The authors write in the introduction that there are only few possibilities regarding statistical software with ready-to-use functions for interval-censored data. This gave me the impression that the five selected approaches were among those few, but this is only partly true. Whereas the models could be fitted using existing SAS routines, the standard errors for percentiles obviously had to be computed/programmed by hand, and R plotting functions were used (do such possibilities not exist in SAS?) which makes the recommendations by far more difficult to use for non-statistician readers. The authors should maybe make it clearer in the introduction that their paper also does not provide an easy solution to the software problem.

5. Again closely related to the former point: The authors only use SAS. Can they please provide more information about functions in other software packages such as R and STATA, so that readers who don't have access to SAS can still use their recommendations?

6. On page 15 the authors explain the approach for interval censoring and how the values for L and R are chosen in the different groups. For group P3 they say that R is chosen to be infinity. I wonder if that is equivalent to treating those values as right-censored or, respectively, how this works technically/computationally or what would change if some real, albeit very large, value would be used instead. Can the authors elaborate a bit more?

7. Would it be possible to integrate covariates in the models? And as the authors seem to think that the percentiles are of particular importance: Did they think about using some quantile regression model? I don't want the authors to actually use these models, but a short discussion would be informative.

Minor points:

1. It would be helpful to have an explanatory graph at the beginning that shows the important time points (such as hospital discharge and the main timepoints used in the study), maybe along with the different types of censoring and trunkation. It would help to understand the situation and in consequence, parts of the explanations later on could be shortened a bit.

2. On page 15 (top) the authors write that there is both left- and right-censoring present and then write "Therefore, the EBOV persistence data can be rightly be considered as interval-censored." This conclusion is wrong. Interval-censoring is not defined this way, but rather that the event of interest happens between two fixed time points. Actually, the authors continue with the correct definition, so I recommend to either delete or rewrite this sentence.

3. The authors could consider moving Figure 3 to the appendix. It is already pretty clear from looking at Figure 2 that the three interval-censoring approaches yield comparable results and I therefore expected the confidence intervals to overlap. So there is not much additional information.

4. Is there a particular reason for choosing the 50th, 75th and 90th percentiles? Any clinical interpretation? If so, a short explanation would be nice.

6. PLOS authors have the option to publish the peer review history of their article (what does this mean?). If published, this will include your full peer review and any attached files.

Reviewer #1: No

Reviewer #2: No

---

## [Author Response · Author response to Decision Letter 0]

8 Sep 2021

We are also pleased to respond to the very important reviewers’ comments as follows:

Reviewer #1: The authors of this manuscript present convincing evidence for use of statistical methods that consider interval censoring when examining persistence of EBOV in semen among SLEVPS participants. Their methods section does a good job introducing different statistical options along with their weaknesses. However, the writing in other sections of this paper, while grammatically correct, is often not written with clear arguments and supporting evidence. The introduction and discussion sections suffer in particular. It is unclear whether the authors are writing a methods tutorial paper with SLEVPS as a convenient example, if they are writing specifically to researchers who work with data like SLEVPS, or if they are trying to improve estimates of EBOV persistence in semen. I recommend that the authors think about what they are trying to communicate specifically with this manuscript and to whom. Once that is identified, I think it will be easier to resolve many of the issues. Full comments in attached file.

AUTHORS RESPONSE: we are writing specifically to researchers who work on persistence estimation of Ebola in body fluids, like is the case for the SLEVPS, however these methodologies may be applicable to other areas facing similar study design challenges.

GENERAL NOTES

COMMENT 1: Writing style often has a lot of fluff. Could generally be more to the point. 

AUTHORS RESPONSE: Manuscript has been revised to maximize clarity.

COMMENT 2: Authors have a tendency to make vague claims with little support. 

AUTHORS RESPONSE: We have revised to ensure that claims are supported by evidence.

COMMENT 3: I expected some examination of existing literature (or of mention of lack thereof) that has either used or failed to use appropriate statistical approaches to evaluating time to event data in this context. However, that context is not present in this manuscript. What are the mistakes people are making? How might this have affected previous estimation of EBOV persistence in semen? What do the authors findings mean in the broader context of EBOV persistence research? 

AUTHORS RESPONSE: We have added examination of literature in the introduction section lines 142-157. Many of the Ebola persistence studies are small in size and a good number involve case studies that are mostly descriptive. reporting the maximum duration of persistence. We found only one study (Sissoko et al 2016) that used the time series approach in the determination of persistence, and another (Subtil et al, 2017) that acknowledged the interval censored nature of the semen EBOV persistence data and used the appropriate methods. However no indepth description on how they determined the lower and upper bounds for the event of interest in the context of left- and interval censored events. Our paper seems to be among the first to describe thoroughly all aspects of the study design of EBOV persistence, as well as on how parameters can be estimated both in the context of non-parametric and parametric survival modelling. In the broader context our paper seeks to demonstrate the issues to consider designwise and analysis-wise while estimating persistence and proposes survival analyses techniques that take into account the interval nature of the data to accomodate the three types of censoring. In the presence of censoring, the survival techniques are still the most suited for assessment of persistence of EBOV in semen of survivors.

ABSTRACT

COMMENT 4: The methods themselves are well-described, but insight into the rationale for choosing this particular set of methods may be lacking. 

AUTHORS RESPONSE: We have included rationale on the RC and IC methods on lines 237-242, and for the choice behind the Weibull parametric distribution on 388-394.

COMMENT 5: [256-261] This explanation is correct but the equation is incorrect. The midpoint between l2 (time of last + visit) and t2 (time of first - visit) would be (*). This error also occurs in Table 1.

AUTHORS RESPONSE: " Counting from time 0, the time (*) in its most simplified form. To conserve space, we have decided to keep the simplified original form of the equation. (Please note (*) is a mathematical equation which could not be printed in this textbox. Full equations are provided in the letter to reviewers (pdf format).

COMMENT 6: [265-267] There is a solid statistical rationale for why midpoint imputation can underestimate the standard error. Bias when using midpoint imputation also has a solid foundation, but the consequences are more conceptually complex. I think the way this is reported is fine but would prefer to see these two biases discussed as separate ideas rather than in one long sentence. 

AUTHORS RESPONSE: We have separated these into two statements (lines 308-311)

COMMENT 7: [ [307] Would prefer to see NPLME KM written out. Appears in a heading and what it means is not written out before this point. 

AUTHORS RESPONSE: the NPMLE has been written out in full on line 278. We have also written in out in full in the heading (line 349-351)

COMMENT 8: [ [308-309] Are these actually recent developments? The original Turnbull paper was published in the 70s and a quick google search shows that an R package accommodating these methods was published around 2010. Avoid making unsubstantiated or unclear claims. 

AUTHORS RESPONSE: We have omitted the word "recent" from the statement.

COMMENT 9: [ [307-341, 368-386] While I am not opposed to more equation-based explanations in papers, the choice to provide this level of detail in these sections is somewhat puzzling. These seem like opportunities to explain the why these approaches may be preferred and conceptually how they work. Perhaps the approach used in these sections confuses me because it is unclear what audience the authors are intending to reach. 

AUTHORS RESPONSE: We still think it is important to convey this information to readers so as to also enable those with interest in understanding the theoretical background as well as, in other cases to get an idea as to how the estimates were derived from the formulae.

DISCUSSION

COMMENT 10: [450-451] One cannot say that Approach 4 is preferred simply because its estimates are "more precise and statistically unbiased." Inference is limited to the data at hand in this project. Thus, one cannot know the true unbiased estimate, and increased precision doesn't necessarily indicate an improvement. An example of this issue – midpoint imputation has very precise estimates at 50% and 75% -- is already baked into this manuscript. The authors need to consider and articulate their arguments clearly. They have outlined reasons why we would expect approaches 1-3 to generate estimates that poorly resemble the truth, and how approaches 4-5 account for the limitations of 1-3. Lean on that argument here. 

AUTHORS RESPONSE: We have revised text accordingly.

COMMENT 11: [454-458] Underestimated relative to what? 

AUTHORS RESPONSE: We have revised text. Relative to KM-IC curve (lines 519-520)

 

Reviewer #2: The authors present an interesting paper that describes different approaches to analysis of time to event data subject to different kinds of censoring. This endeavour deserves merit, and they explain very nicely the different approaches and show their results using a data set with information about traces of the Ebola virus in semen.

AUTHORS RESPONSE: We thank the reviewer for this motivating comment

MAJOR ISSUES

COMMENT 1: It is unclear what the main aim of using these models is. Should individual out-of-sample predictions be possible in order to advise future Ebola survivors? Or does the main interest lie in observing the mean or median behaviour (or the percentiles) of the population of Ebola survivors? 

AUTHORS RESPONSE: The main interest is in describing the performance of these models, and commenting on the differences in the estimation of the cumulaive rate of persistence distribution as well as and percentile estimates using SLEVPS sample data.

COMMENT 2: This point is closely related to the former point and actually the most important part of my review: The authors can only describe the different estimated survival curves/percentiles/standard errors. But it is impossible to say which of the models is best, because the authors fail to define criteria which can be used to measure the model quality. At several places the authors claim that one or the other model is unbiased, but they cannot say this without a criterion for model quality. They sometimes describe that a model leads to more precise estimates in the sense of having smaller confidence intervals, but this does not necessarily mean that the respective model is correct. 

AUTHORS RESPONSE: We have revised manuscript to compare the strength and the limitation of each of the approaches to estimating persistence.

COMMENT 3: Depending on the main goal of fitting these models (see point 1), I suggest the following changes: If the main goal is prediction, the authors should predefine criteria for predictive model choice such as proper scoring rules (e.g. the Brier score) that can take into account both sharpness (precision) and calibration of a predictive model or judge the calibration alone. In addition, they should separate the data set in two parts, the training set and the test set, because otherwise the model fit will always be too good. Alternatively, they could use some kind of cross-validation.

If, however, the main aim of the models is parameter estimation/a description of the population, they should show some criteria for model fit/model choice such as some version of AIC/BIC/DIC or the likes that are appropriate for the models they use and allow for a somewhat independent comparison of the quality of the models. In addition, they could add a plot that shows the observed data, so that it is easier to judge.

If the authors don't want or cannot do this, they could alternatively give a restructured description of the different approaches and their advantages and disadvantages (e.g. distributional assumptions, number of parameters,...) and only show the results of the models as an addition at the and, but they should remove all judgement from the results part that is not justified (all about bias or the notion that a more precise confidence interval is always better).

AUTHORS RESPONSE: We have followed this latter suggestion to discuss more on how the five survival models selected function, the results of the fit and more on advantages and disadvantages. We have only discussed on the perfomance of the models based on the estimates they give.

COMMENT 4: It becomes already clear from the description of the approaches that approach 1 and 2 that take only right-censoring into account are not appropriate for the data set. Therefore it is not at all surprising that the estimates are quite far away from the ones from the models for interval-censored data, this statement is a bit trivial. It is still nice to see, but it does not add much to the paper, while using relatively much space in the methods part. I recommend to shorten the explanation a bit and focus more on the other three methods. 

AUTHORS RESPONSE: We have revised the reduced the description of the right-censored data and add more on the interval censored methods (lines 388-410)

COMMENT 5:The authors write in the introduction that there are only few possibilities regarding statistical software with ready-to-use functions for interval-censored data. This gave me the impression that the five selected approaches were among those few, but this is only partly true. Whereas the models could be fitted using existing SAS routines, the standard errors for percentiles obviously had to be computed/programmed by hand, and R plotting functions were used (do such possibilities not exist in SAS?) which makes the recommendations by far more difficult to use for non-statistician readers. The authors should maybe make it clearer in the introduction that their paper also does not provide an easy solution to the software problem. 

AUTHORS RESPONSE: Thanks for highlighting this. We have included this in the introduction (lines 137-141)

COMMENT 6: Again closely related to the former point: The authors only use SAS. Can they please provide more information about functions in other software packages such as R and STATA, so that readers who don't have access to SAS can still use their recommendations? 

AUTHORS RESPONSE: Yes. We have included R and STATA as among the statistical software programs that can also estimate IC survival data both in non-parametric and parametric way. (lines 408-410; 566-570) and integration of covariates in the IC regression models lines (561-571).

COMMENT 7: On page 15 the authors explain the approach for interval censoring and how the values for L and R are chosen in the different groups. For group P3 they say that R is chosen to be infinity. I wonder if that is equivalent to treating those values as right-censored or, respectively, how this works technically/computationally or what would change if some real, albeit very large, value would be used instead. Can the authors elaborate a bit more? 

AUTHORS RESPONSE: We have added a sentence clarifying that for the right-censored individual the upper limit of 'infinity' is usually replaced by a missing value, which the program interprets it as right-censored value. (lines 328-330)

COMMENT 8: Would it be possible to integrate covariates in the models? And as the authors seem to think that the percentiles are of particular importance: Did they think about using some quantile regression model? I don't want the authors to actually use these models, but a short discussion would be informative. 

AUTHORS RESPONSE: Yes it is possible to integrate covariates in the semi-parametric PH IC models. We have brief paragraph that explains this and different statistical packages that can analyze this, also in the context of parametric modelling (lines 561-570)

MINOR POINTS

COMMENT 9: It would be helpful to have an explanatory graph at the beginning that shows the important time points (such as hospital discharge and the main timepoints used in the study), maybe along with the different types of censoring and trunkation. It would help to understand the situation and in consequence, parts of the explanations later on could be shortened a bit. 

AUTHORS RESPONSE: Ythe important timepoints have been described in Figure 1 that was saved separately as S1 Fig.tiff. 

 COMMENT 10: On page 15 (top) the authors write that there is both left- and right-censoring present and then write "Therefore, the EBOV persistence data can be rightly be considered as interval-censored." This conclusion is wrong. Interval-censoring is not defined this way, but rather that the event of interest happens between two fixed time points. Actually, the authors continue with the correct definition, so I recommend to either delete or rewrite this sentence.

AUTHORS RESPONSE: We have revised lines 323-331 adequately describes the interval censoring. 

COMMENT 11: The authors could consider moving Figure 3 to the appendix. It is already pretty clear from looking at Figure 2 that the three interval-censoring approaches yield comparable results and I therefore expected the confidence intervals to overlap. So there is not much additional information. 

AUTHORS RESPONSE: We have moved Figure 3 to the appendix

COMMENT 12: Is there a particular reason for choosing the 50th, 75th and 90th percentiles? Any clinical interpretation? If so, a short explanation would be nice. 

AUTHORS RESPONSE: Percentiles of viral persistence in semen, provides the probability of the virus EBOV persisting beyond a certain time period, this being of clinical and public health importance as it helps in informing semen testing survivor programmes; as well as in formulating policies surrounding duration of use of certain preventive measures including sexual abstinence and condom use aimed at minimizing sexual transmission of the virus and therefore preventing possibility future outbreaks. Extreme upper tail viral persistence percentiles furthermore are important in understanding how long it takes following ETU discharge that a group of survivors slowest to clear EBOV become EBOV-free. We have added this explanation on lines 571-578.

We thank both reviewers for their time and for the very useful comments they provided. 

We are kindly resubmitting our manuscript for consideration.

Sincerely,

Ndema Habib, PhD

Statistician, The Department of Sexual and Reproductive Health Research (SRH), WHO, Geneva.

---

## [Decision Letter · Decision Letter 1]

27 Sep 2021

PONE-D-21-05351R1Statistical methodologies for evaluation of the rate of persistence of Ebola virus in semen of male survivors in Sierra LeonePLOS ONE

Dear Dr. Habib,

Thank you for submitting your manuscript to PLOS ONE. After careful consideration, we feel that it has merit but does not fully meet PLOS ONE’s publication criteria as it currently stands. Therefore, we invite you to submit a revised version of the manuscript that addresses the points raised during the review process.

We look forward to receiving your revised manuscript.

Kind regards,

Mohammad Asghari Jafarabadi

Academic Editor

PLOS ONE

Journal Requirements:

Reviewers' comments:

Reviewer's Responses to Questions

**Comments to the Author**

1. If the authors have adequately addressed your comments raised in a previous round of review and you feel that this manuscript is now acceptable for publication, you may indicate that here to bypass the “Comments to the Author” section, enter your conflict of interest statement in the “Confidential to Editor” section, and submit your "Accept" recommendation.

Reviewer #1: All comments have been addressed

Reviewer #2: (No Response)

2. Is the manuscript technically sound, and do the data support the conclusions?

Reviewer #1: Yes

Reviewer #2: Yes

3. Has the statistical analysis been performed appropriately and rigorously? 

Reviewer #1: Yes

Reviewer #2: Yes

4. Have the authors made all data underlying the findings in their manuscript fully available?

Reviewer #1: No

Reviewer #2: No

5. Is the manuscript presented in an intelligible fashion and written in standard English?

Reviewer #1: Yes

Reviewer #2: Yes

6. Review Comments to the Author

Reviewer #1: Thank you for so graciously considering and responding to our many comments. The revised manuscript is much improved and does a good job presenting the already solid research done by the authors.

There are a few awkward sentences and grammatical errors (comma splices and run-on sentences are a common problem) at: 70-73, 76-79, 79-81, 82-85, 88-91, 92-93 & potentially other locations.

[126-132] Side note (no changes/response requested) – left censoring is actually a big problem in many public health studies because of immortal time bias.

[126-141] Good paragraph

[142-157] Thank you for adding, this is helpful context.

[Table 1] Unfortunately, equations in author response (and from my initial comment) were turned into an asterisk by (I suspect) the program that created my pdf, so I cannot fully evaluate author's response.

[530-534] It's interesting thinking about this problem with such a high proportion of left-censored participants.

Reviewer #2: Thank you for addressing my comments. I especially like the software information that is now given at several places.

My only recommendation is to go again through the manuscript with particular attention to grammar mistakes/funny expressions. Especially in the new or changed parts of the manuscript, there are several small mistakes, probably due to some haste in which the manuscript was rewritten.

7. PLOS authors have the option to publish the peer review history of their article (what does this mean?). If published, this will include your full peer review and any attached files.

Reviewer #1: No

Reviewer #2: No

---

## [Author Response · Author response to Decision Letter 1]

5 Nov 2021

We are also pleased to respond to the very important reviewers’ comments as follows:

Reviewer #1: Review Comments to the Author

Thank you for so graciously considering and responding to our many comments. The revised manuscript is much improved and does a good job presenting the already solid research done by the authors.

AUTHORS RESPONSE: We appreciate your comment.

There are a few awkward sentences and grammatical errors (comma splices and run-on sentences are a common problem) at: 70-73, 76-79, 79-81, 82-85, 88-91, 92-93 & potentially other locations.

AUTHORS RESPONSE: Thank you. We have revised and corrected as per lines 71-74, 77-82, 83-86, 89-91, 92-93, as well as in other places.

[126-132] Side note (no changes/response requested) – left censoring is actually a big problem in many public health studies because of immortal time bias.

AUTHORS RESPONSE: This is well noted.

[126-141] Good paragraph

AUTHORS RESPONSE: Thank you for encouraging remark.

[142-157] Thank you for adding, this is helpful context.

AUTHORS RESPONSE: Noted. Thank you,

[Table 1] Unfortunately, equations in author response (and from my initial comment) were turned into an asterisk by (I suspect) the program that created my pdf, so I cannot fully evaluate author's response.

AUTHORS RESPONSE: In our last Response to reviewers’ letter, we prepared and attached separately authors’ responses in Word format as well. There, all the equations appear as they should have. 

[530-534] It's interesting thinking about this problem with such a high proportion of left-censored participants.

AUTHORS RESPONSE: We thank the reviewer for this motivating comment

Reviewer #2: Review Comments to the Author

Thank you for addressing my comments. I especially like the software information that is now given at several places.

My only recommendation is to go again through the manuscript with particular attention to grammar mistakes/funny expressions. Especially in the new or changed parts of the manuscript, there are several small mistakes, probably due to some haste in which the manuscript was rewritten.

AUTHORS RESPONSE: Thank you. We have now revised and corrected the grammatical errors.

We thank both reviewers for their time and for the very useful comments they provided. 

Sincerely,

Ndema Habib, PhD

Statistician, The Department of Sexual and Reproductive Health Research (SRH), WHO, Geneva.

---

## [Decision Letter · Decision Letter 2]

3 Feb 2022

PONE-D-21-05351R2Statistical methodologies for evaluation of the rate of persistence of Ebola virus in semen of male survivors in Sierra LeonePLOS ONE

Dear Dr. Habib,

Thank you for submitting your manuscript to PLOS ONE. After careful consideration, we feel that it has merit but does not fully meet PLOS ONE’s publication criteria as it currently stands. Therefore, we invite you to submit a revised version of the manuscript that addresses the points raised during the review process.

We look forward to receiving your revised manuscript.

Kind regards,

Mohammad Asghari Jafarabadi

Academic Editor

PLOS ONE

Reviewers' comments:

Reviewer's Responses to Questions

**Comments to the Author**

1. If the authors have adequately addressed your comments raised in a previous round of review and you feel that this manuscript is now acceptable for publication, you may indicate that here to bypass the “Comments to the Author” section, enter your conflict of interest statement in the “Confidential to Editor” section, and submit your "Accept" recommendation.

Reviewer #2: All comments have been addressed

Reviewer #3: (No Response)

2. Is the manuscript technically sound, and do the data support the conclusions?

Reviewer #2: Yes

Reviewer #3: No

3. Has the statistical analysis been performed appropriately and rigorously? 

Reviewer #2: Yes

Reviewer #3: No

4. Have the authors made all data underlying the findings in their manuscript fully available?

Reviewer #2: No

Reviewer #3: No

5. Is the manuscript presented in an intelligible fashion and written in standard English?

Reviewer #2: Yes

Reviewer #3: Yes

6. Review Comments to the Author

Reviewer #2: (No Response)

Reviewer #3: Comments:

As I understand the authors want to study the survival period of EBOV (Ebola virus) in the semen of patients discharged from the hospital. In other words, it can be stated that the probability of occurrence of event X (X is negative EBOV) with the available censored data.

1. The total sample size of the study is only 203 cases and out of 203, 88 cases experienced the event (negative test of EBOV) before the day of recruitment. Only 115 cases were included for follow up study. At the same time follow up period was comparatively long i.e. till the last case was tested negative or withdrawn from the study; about 451 - 540 days. After 270 days, only 6 cases were followed for such a long period of another 270 days. The follow-up period could be ended after 250 days. As the remaining sample size for the tail period of the curve was much smaller, may not be adequate to reflect the probability distribution.

2. Overall, the sample size is not sufficient to find the survival or time to event model in the given situation. As it plays an important role in testing the hypotheses regarding the suitability of methods. In such a situation, the alternative approach is to carry out the simulation studies with varying sample sizes and test the consistency of the outcomes or results.

3. The authors did not explore time-dependent (or non-proportional hazard) scenarios in this study as the survival time distribution for non-proportional time survival will be different.

4. Authors should check the work by Gruttola and Lagakos (1989) who proposed nonparametric and weakly structured parametric methods for analysing survival data in which both the time origin and the failure event can be right- or interval-censored? Full detail can be found at https://www.jstor.org/stable/pdf /2532030.pdf

5. Interval-censored data can be easily confused with grouped survival data. However, there is actually a fundamental difference between these two data structures although both usually appear in the form of intervals. The grouped survival data can be seen as a special case of interval-censored data and commonly mean that the intervals for any two subjects either are completely identical or have no overlapping. In contrast, the intervals for interval-censored data may overlap in any way. Because of this structural difference, statistical methods for grouped survival data are much more straightforward than those for interval-censored data. Did authors check the structure of their data and need to be clarified in the methods section?

6. The survival model to be used for such data are known and discussed below. It may help the author to prepare the data file for analysis.

7. 203 Ebola patients (who had been discharged from the hospital or received treatment and confirmed from the hospital record) were recruited in the study. It may not be correct if we take the start time as the day of recruitment in the study. The starting time or initial time of presence of EBOV as zero-day to be taken as the date of discharge from the hospital expecting on that day the EBOV virus is present in the semen. Accordingly, the day of recruitment is to be taken as the first observation for follow up of cases and the number of days to be counted from discharge day to the recruitment day as Ti (Time) for all the 203 patients. Ti should not be taken as zero, it may vary for each patient/recruited case depending on the date of discharge and day of recruitment ( in days and condition i.e. date of discharge < day of recruitment) or to be taken as a result of the first day of follow up.

8. Censored (left/right)data ( δ): In this scenario, we have both left and right-censored data at a different level

i) From the day of discharge to the day of recruitments for each patient, if the EBOV positive ( δ = 0) else left-censored ( δ = -1).

ii) During the follow up of all the remaining positive cases, for each case as when tested negative ( δ = 1) i.e. right-censored and no follow up. But if positive, but lost to follow up or withdrawn at that scheduled date of lost to follow up test to be taken with time of even(T) censored value(( δ = 0), and so on.

iii) After preparing the data file, the survival model can be used to calculate the survival probability of EBOV. There are several models, but the choice model will be parametric or non-parametric. But the sample size may not be adequate to use the parametric model. However, based on the characteristics of survival data one can use and find a suitable model. The detailed procedure is available in the book noted below for reference.

9. There are three or four methods/approaches suggested in this study for fitting the survival model which may be misleading or confusing the reader as the model should be applicable to any such data or possible generalization.

Book Name: Statistics for Biology and Health: Survival Analysis

Techniques for Censored and Truncated data

By John P. Klein & Melvin L. Moeschberger

Kindly read Chapter 3 also to understand the Censoring and Truncation especially page 56 to 66.

7. PLOS authors have the option to publish the peer review history of their article (what does this mean?). If published, this will include your full peer review and any attached files.

Reviewer #2: No

Reviewer #3: No

---

## [Author Response · Author response to Decision Letter 2]

27 Jul 2022

Title: Statistical methodologies for evaluation of the rate of persistence of Ebola virus in semen of male survivors in Sierra Leone

Comments:

As I understand the authors want to study the survival period of EBOV (Ebola virus) in the semen of patients discharged from the hospital. In other words, it can be stated that the probability of occurrence of event X (X is negative EBOV) with the available censored data.

1. The total sample size of the study is only 203 cases and out of 203, 88 cases experienced the event (negative test of EBOV) before the day of recruitment. Only 115 cases were included for follow up study. At the same time follow up period was comparatively long i.e. till the last case was tested negative or withdrawn from the study; about 451 - 540 days. After 270 days, only 6 cases were followed for such a long period of another 270 days. The follow-up period could be ended after 250 days. As the remaining sample size for the tail period of the curve was much smaller, may not be adequate to reflect the probability distribution. 

AUTHORS RESPONSE: For the interval censored survival estimation of EBOV persistence in semen, all the 203 male participants were followed up to the time they tested consecutively twice negative for EBOV in semen. This implies that even the 88 participants who had already experienced the event had contributed persistence data at enrolment and also at one additional post-enrolment consecutive visit; with the right-limit of the interval censoring interval set as the first of the two consecutively negative visits i.e the enrolment visit. SLEVPS team was interested also in the estimation of EBOV persistence in other body fluids, therefore all the 203 participants continued to contribute persistence data for other body fluids, even after being EBOV-free in semen. Understanding how long the persistence of the virus in semen lasts is of public health importance and this is behind using all available information on persistence even for those whose virus in the semen persisted the longest.(WHO/VHF/2018.1). We have furthermore estimated the percentiles of persistence whose figures are unaffected by the extreme observations.

2. Overall, the sample size is not sufficient to find the survival or time to event model in the given situation. As it plays an important role in testing the hypotheses regarding the suitability of methods. In such a situation, the alternative approach is to carry out the simulation studies with varying sample sizes and test the consistency of the outcomes or results. 

AUTHORS RESPONSE: Thanks for your suggestion. The aim of the SLEVPS was to estimate the duration and rate at which the EBOV persists in semen. All recruited subjects were followed up to the time of confirmed EBOV-negative in semen); or until the censoring time (premature discontinuation or loss to follow up), counted from time of discharge from Ebola Treatment Unit (ETU). Given the nature of this outcome, we believe strongly that the survival (time-to-event) methods would be the most appropriate for this evaluation. This study was designed and implemented during an emergency setting of the EVD outbreak with the sample size selected based on convenience.The SLEVPS researchers had no control over the number of participants to recruit nor how soon to recruit following recovery (as described in the manuscript). For this paper we describe how the traditional right-censored methods and the interval-censored methods could be used and how they perform in the evaluation of EBOV persistence when massive left-censoring is present, due to the limitations in study design for Ebola persistence. We agree with the reviewer on the importance of conducting simulations, however for the current manuscript we believe that the data and results we presented in our manuscript still provide interesting insights to dealing with data of this nature. 

3. The authors did not explore time-dependent (or non-proportional hazard) scenarios in this study as the survival time distribution for non-proportional time survival will be different. 

AUTHORS RESPONSE: In the current paper we did not explore non-proportional hazards assumptions since it was not aimed at comparing survival (or hazard rates) between exposure groups. The aim of this manuscript was solely to estimate persistence (survival) among male Ebola disease survivors, using the right- and the interval-censored methods and describe their distributions. 

4. Authors should check the work by Gruttola and Lagakos (1989) who proposed nonparametric and weakly structured parametric methods for analysing survival data in which both the time origin and the failure event can be right- or interval-censored? Full detail can be found at https://www.jstor.org/stable/pdf /2532030.pdf 

AUTHORS RESPONSE: Thanks for your suggestion and for the reference. We have looked at it. In our paper we have illustrated hthe use of both nonparametric and parametric (Weibull) interval censored models could be applied to interval- and right.-censored Ebola persistence data. 

5. Interval-censored data can be easily confused with grouped survival data. However, there is actually a fundamental difference between these two data structures although both usually appear in the form of intervals. The grouped survival data can be seen as a special case of interval-censored data and commonly mean that the intervals for any two subjects either are completely identical or have no overlapping. In contrast, the intervals for interval-censored data may overlap in any way. Because of this structural difference, statistical methods for grouped survival data are much more straightforward than those for interval-censored data. Did authors check the structure of their data and need to be clarified in the methods section? 

AUTHORS RESPONSE: Our data is not grouped sutvival data, rather it was data that was collected on individual males survivors, with a known time of origin (date of ETU discharge) and known time interval in which the event of primary interest (confirmed EBOV clearance from semen) or censoring time, for each participant. This is already clarified in the manuscript (lines 211-235).

6. The survival model to be used for such data are known and discussed below. It may help the author to prepare the data file for analysis.

AUTHORS RESPONSE: Thanks for this comment. The data file was already prepared and was used to run the proposed models and estimates. 

 

7. 203 Ebola patients (who had been discharged from the hospital or received treatment and confirmed from the hospital record) were recruited in the study. It may not be correct if we take the start time as the day of recruitment in the study. The starting time or initial time of presence of EBOV as zero-day to be taken as the date of discharge from the hospital expecting on that day the EBOV virus is present in the semen. Accordingly, the day of recruitment is to be taken as the first observation for follow up of cases and the number of days to be counted from discharge day to the recruitment day as Ti (Time) for all the 203 patients. Ti should not be taken as zero, it may vary for each patient/recruited case depending on the date of discharge and day of recruitment ( in days and condition i.e. date of discharge < day of recruitment) or to be taken as a result of the first day of follow up.

AUTHORS RESPONSE: Indeed we have taken Time 0 as the date of ETU discharge, and not the date of recruitment. Recruitment visit is the first day of post-ETU discharge follow-up.

8. Censored (left/right)data ( δ): In this scenario, we have both left and right-censored data at a different level 

i) From the day of discharge to the day of recruitments for each patient, if the EBOV positive ( δ = 0) else left-censored ( δ = -1).

ii) During the follow up of all the remaining positive cases, for each case as when tested negative ( δ = 1) i.e. right-censored and no follow up. But if positive, but lost to follow up or withdrawn at that scheduled date of lost to follow up test to be taken with time of even(T) censored value(( δ = 0), and so on.

iii) After preparing the data file, the survival model can be used to calculate the survival probability of EBOV. There are several models, but the choice model will be parametric or non-parametric. But the sample size may not be adequate to use the parametric model. However, based on the characteristics of survival data one can use and find a suitable model. The detailed procedure is available in the book noted below for reference. 

AUTHORS RESPONSE: This is noted, and It is along the lines that we have described the models proposed.

9. There are three or four methods/approaches suggested in this study for fitting the survival model which may be misleading or confusing the reader as the model should be applicable to any such data or possible generalization.

Book Name: Statistics for Biology and Health: Survival Analysis 

Techniques for Censored and Truncated data

By John P. Klein & Melvin L. Moeschberger

Kindly read Chapter 3 also to understand the Censoring and Truncation especially page 56 to 66. 

AUTHORS RESPONSE: This is noted. Thanks for your comments. We have read and referred to this book in our paper already (line 704, reference no. 13)

---

## [Decision Letter · Decision Letter 3]

4 Sep 2022

Statistical methodologies for evaluation of the rate of persistence of Ebola virus in semen of male survivors in Sierra Leone

PONE-D-21-05351R3

Dear Dr. Habib,

We’re pleased to inform you that your manuscript has been judged scientifically suitable for publication and will be formally accepted for publication once it meets all outstanding technical requirements.

Kind regards,

Mohammad Asghari Jafarabadi

Academic Editor

PLOS ONE

Additional Editor Comments (optional):

Reviewers' comments:

Reviewer's Responses to Questions

**Comments to the Author**

1. If the authors have adequately addressed your comments raised in a previous round of review and you feel that this manuscript is now acceptable for publication, you may indicate that here to bypass the “Comments to the Author” section, enter your conflict of interest statement in the “Confidential to Editor” section, and submit your "Accept" recommendation.

Reviewer #3: All comments have been addressed

Reviewer #4: All comments have been addressed

2. Is the manuscript technically sound, and do the data support the conclusions?

Reviewer #3: Yes

Reviewer #4: Yes

3. Has the statistical analysis been performed appropriately and rigorously? 

Reviewer #3: Yes

Reviewer #4: Yes

4. Have the authors made all data underlying the findings in their manuscript fully available?

Reviewer #3: Yes

Reviewer #4: Yes

5. Is the manuscript presented in an intelligible fashion and written in standard English?

Reviewer #3: Yes

Reviewer #4: Yes

6. Review Comments to the Author

Reviewer #3: Thanks to all the authors for giving due importance to each comments and revising the manuscript accordingly.

Reviewer #4: In my opinion, the authors have correctly answered the questions and comments of the third reviewer.

Furthermore, given that the researchers have not considered the investigated data merely as a practical example for the investigated survival analysis methods and they have emphasized the health and medical aspect of the investigated data in the title, abstract, and introduction of the article, it is better to report the rate of persistence of Ebola virus in semen of male survivors obtained from the better models in the result and conclusion section of the abstract.

7. PLOS authors have the option to publish the peer review history of their article (what does this mean?). If published, this will include your full peer review and any attached files.

Reviewer #3: No

Reviewer #4: No

---

## [Editor Report · Acceptance letter]

11 Sep 2022

PONE-D-21-05351R3 

Statistical methodologies for evaluation of the rate of persistence of Ebola virus in semen of male survivors in Sierra Leone 

Dear Dr. Habib:

I'm pleased to inform you that your manuscript has been deemed suitable for publication in PLOS ONE. Congratulations! Your manuscript is now with our production department. 

Kind regards, 

on behalf of

Professor Mohammad Asghari Jafarabadi 

Academic Editor

PLOS ONE